# Coupled pharmacokinetic model unveils drug-drug interactions in plasma concentration

Hong Huang[1☯], Chaoyang Li[2☯], Qianqian Chen[3], Chumeng Zhuang[3], Li Yu[3,4], Weifeng Jin[5*], Xiaohong Li [5*]

**1** The School of Humanities and Management, Zhejiang Chinese Medical University, Hangzhou, China, **2** The Fourth School of Clinical Medicine, Zhejiang Chinese Medical University, Hangzhou, China, **3** School of Basic Medical Sciences, Zhejiang Chinese Medical University, Hangzhou, China, **4** Key Laboratory of Drug Safety Evaluation and Research of Zhejiang Province, Center of Safety Evaluation and Research, Hangzhou Medical College, Hangzhou, China, **5** School of Pharmaceutical Sciences, Zhejiang Chinese Medical University, Hangzhou, China

☯ The first two authors contributed equally to this work.
* jin_weifeng@126.com (WFJ); li_xiaoh2005@163.com (XHL)

## Abstract

In oral drug pharmacokinetics (PK), drug-drug interactions are inevitable, yet traditional compartmental models struggle to effectively quantify such processes. This study proposes a linearly coupled two-compartment PK model, where the coupling term is defined as a linear function of another drug's amount to strike a balance between model simplicity and physiological interpretability. The model introduces parameter heterogeneity and linear interaction terms based on the classical compartmental structure, more accurately capturing concentration-dependent dynamic changes during combined drug administration. To address the model's nonlinear characteristics and high-dimensional parameters, a hierarchical optimization numerical solution algorithm was developed, enhancing computational efficiency while validating robustness against Gaussian noise. Through systematic analysis of key PK metrics ($C_{max}$, $T_{max}$, $AUC$, and $t_{1/2}$), the study reveals the mechanisms by which absorption and clearance parameter variations influence drug distribution in vivo. Combining numerical simulations, parameter ablation experiments, and real-world data validation, the full model (retaining all linear interaction terms) outperforms the simplified model in both goodness-of-fit and information criteria, demonstrating superior interpretability and predictive performance. Overall, this model offers an intermediate solution between traditional compartmental models and PBPK models, providing a novel methodological framework for quantitative research on drug-drug interactions.

**Data availability statement:** All relevant data are within the manuscript and its Supporting Information files.

**Funding:** This research was supported by the Zhejiang Natural Science Foundation (grant numbers LZYQ25H270001 to Y.L.; LY24H270007 to J.W.F.; LY22H270002 to Y.L.), the Zhejiang Postdoctoral Research Foundation (grant number ZJ2023021 to Y.L.), and the National Natural Science Foundation of China (grant number 81904083 to Y.L.).

**Competing interests:** NO authors have competing interests.

## Introduction

Pharmacokinetics (PK) plays an important part in both preclinical and clinical drug studies by quantifying the dynamic behavior of drugs within the organism, which can elucidate their efficacy and safety [1,2]. Data analysis is an essential aspect of PK studies [3]. Typically, PK modeling is under the framework of the compartmental model that utilizes differential equations and the law of amount conservation [4,5] to effectively characterize temporal changes in the amount or concentration of a single drug. For combination drugs, existing PK models assume that the parameters of each component are uniformly affected by the other, which means that the PK model of each component is independent and its parameters are all constants.

In order to maximize effectiveness and reduce side effects, the treatment of chronic illnesses usually depends on the synergistic effects of several oral drugs [6,7]. By utilizing medications with complimentary modes of action, for example, benazepril and amlodipine are commonly used together to treat critical hypertension in an effort to increase antihypertensive efficacy and reduce risk [8]. In addition, glimepiride and metformin together are often used to treat type 2 diabetes [9], which improves the medication's ability to lower blood sugar levels but also raises the possibility of side effects such as hypoglycemia [10]. Therefore, in order to reduce the possibility of adverse reactions [11], it is necessary to quantitatively describe how the two drugs are absorbed, distributed, and eliminated by the body by using PK techniques.

PK modeling of drug combinations typically employs compartmental models, where traditional compartmental models often assume that pharmacokinetic parameters (e.g., clearance, volume of distribution) remain constant over time and drug concentration to simplify model structure and facilitate parameter estimation [12–14]. However, this assumption struggles to capture the dynamic dependencies arising from parameter changes over time or concentration. In recent years, physiologically based pharmacokinetic (PBPK) models have demonstrated some capability to describe such time-varying and interactive properties, yet they often rely on extensive prior information and complex computational frameworks [15].

Based on the pharmacological background and experimental evidence mentioned above, the coupled PK model proposed in this study introduces interaction terms into the concise structure of the classical compartmental model to capture the dynamic coupling effects of drug concentration dependence, thereby achieving a balance between model complexity and explanatory power, providing an intermediate solution between traditional compartmental models and PBPK models for drug interaction research.

The paper structure is as follows: In the Coupled PK Modeling section, a coupled PK model was established, and the influence of model parameters and their effects on PK parameters ($C_{max}$, $T_{max}$, $AUC$, and $t_{1/2}$) were demonstrated through numerical simulations. In the Numerical Solution Method section, a hierarchical optimization method was introduced and validated through four numerical examples. The model's performance analysis and ablation experiments were also conducted to verify its robustness. In the Model Application section, the application of models and

optimization techniques in real systems is provided, demonstrating the feasibility and scientific effectiveness of the models and algorithms. Finally, the model and solving techniques were discussed, and future research directions were outlined.

## Coupled PK modeling

### Mathematical model

In this study, the absorption, distribution, and elimination rates of one drug are considered to be influenced by monotonic changes in the amount of the other drug. In addition, this study is based on an important assumption that it is only applicable to specific dose ranges. Similar to treating extremely small segments of a curve as straight lines when calculating curve integrals,the heterogeneity of pharmacokinetic model parameters is defined as a linear relationship, with the sign of the slope indicating their interaction. For clarity, X will be used to represent one drug and Y to represent the other. The compartmental structure of the coupled PK model is illustrated in Fig 1. Based on the assumptions of parameter heterogeneity and linear relationship, the two-compartment coupled PK model is presented as follows,the specific form is shown in Eq 1.

$$
\begin{cases}
\frac{dx_0}{dt} = -(\alpha_{00} + \alpha'_{00}y_0)x_0 \\
\frac{dx_1}{dt} = (\alpha_{00} + \alpha'_{00}y_0)x_0 - (\alpha_{12} + \alpha'_{12}y_1)x_1 + (\alpha_{21} + \alpha'_{21}y_2)x_2 - (\alpha_{10} + \alpha'_{10}y_1)x_1 \\
\frac{dx_2}{dt} = (\alpha_{12} + \alpha'_{12}y_1)x_1 - (\alpha_{21} + \alpha'_{21}y_2)x_2 \\
\frac{dy_0}{dt} = -(\beta_{00} + \beta'_{00}x_0)y_0 \\
\frac{dy_1}{dt} = (\beta_{00} + \beta'_{00}x_0)y_0 - (\beta_{12} + \beta'_{12}x_1)y_1 + (\beta_{21} + \beta'_{21}x_2)y_2 - (\beta_{10} + \beta'_{10}x_1)y_1 \\
\frac{dy_2}{dt} = (\beta_{12} + \beta'_{12}x_1)y_1 - (\beta_{21} + \beta'_{21}x_2)y_2
\end{cases}
$$

$$(1)$$

where

$x_0, y_0$ = amount of drug in gastrointestinal tract compartment (mg);

$x_1, y_1$ =amount of drug in central compartment (mg);

$x_2, y_2$ = amount of drug in peripheral compartment (mg);

$\alpha_{00}, \beta_{00}$ = rate parameter for absorption ($h^{-1}$);

$\alpha_{12}, \beta_{12}$ = rate parameter of drug transfer from central to peripheral compartment ($h^{-1}$);

$\alpha_{21}, \beta_{21}$ = rate parameter of drug transfer from peripheral to central compartment ($h^{-1}$);

$\alpha_{10}, \beta_{10}$ = rate parameter for elimination ($h^{-1}$);

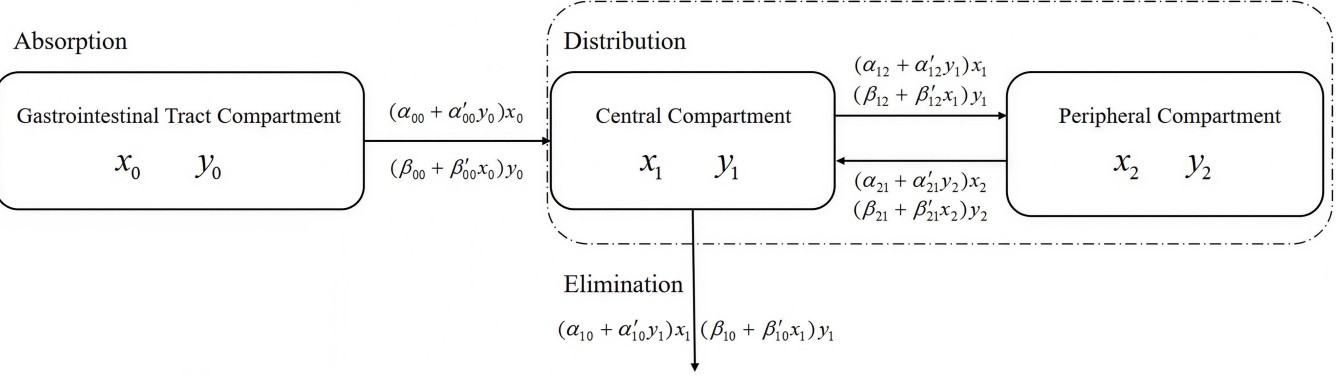

**Fig 1. Illustration of coupled PK model.**

$\alpha'_{00}, \beta'_{00}$ = interaction parameters for absorption ($1 \cdot 10^3$/(mg·h));
$\alpha'_{12}, \beta'_{12}$ = interaction parameter of drug transfer from central to peripheral compartment ($1 \cdot 10^3$/(mg·h));
$\alpha'_{21}, \beta'_{21}$ = interaction parameter of drug transfer from peripheral to central compartment ($1 \cdot 10^3$/(mg·h));
$\alpha'_{10}, \beta'_{10}$ = interaction parameter for elimination ($1 \cdot 10^3$/(mg·h)).

The strength of drug interactions is evaluated through the coupling term by analyzing the sign and magnitude of the interaction parameters. When drug X is administered alone, the parameters $\alpha'_{ij}$ and $\beta'_{ij}$ are zero. As a result, the model is simplified to a traditional deterministic compartmental model.

In cases where both drugs are administered together, a temporal relationship between drug X and drug Y is measured in terms of a parametric linear equation. Its slope is provided to quantity the facilitation or inhibition of drug X by drug Y, and the intercept is used to represent the rate parameter for one drug in the absence of the other. Therefore, the structure of interactions is integrated into the model to extend the conventional PK framework, with a focus on the dynamics of drug-drug interactions in plasma concentrations.

Drug concentrations in plasma, which are commonly measured by tracking plasma samples over time, make up the experimental data for PK models. However, in this paper, the disparity among the volumes of the absorptive, central and peripheral compartments is addressed by converting drug concentration into amount during absorption and transfer processes [16]. The equation used for converting drug plasma concentration to plasma amount is defined by Eq 2,

$$x = Vc \cdot c_x \tag{2}$$

where $x$ is the amount of drug plasma, $Vc$ is the apparent distribution volume of the central compartment, and $c_x$ is the concentration of drug plasma.

The rationality of this formula conversion comes from two aspects. Firstly, the law of conservation of amount states that the amount of a drug is always equal to the product of concentration and volume; The second is the perspective of parameter definition. In pharmacokinetics, the apparent distribution volume $Vc$ does not refer to an actual physical volume, but describes the distribution characteristics of drugs in the body. Specifically, $Vc$ represents that if the drug is evenly distributed within this volume, the drug concentration in the plasma will be equal to the actual measured concentration $c_x$. Since the parameters of the experimental subjects can be assumed to be equal in both single pharmacokinetic and coupled pharmacokinetic programs, the experimental data of the single pharmacokinetic program can be analytically solved [17].

### Numerical simulations

In coupled PK models, some common strategies include enhancing the absorption of drug X in the central compartment and promoting its transfer from the peripheral compartment to increase the amount of drug X in the central compartment. Inhibiting the elimination of drug X and reducing its transfer from the central compartment to the peripheral compartment are also effective strategies. According to the structure of the model, the interaction parameters can be either positive or negative. The negative parameter serves as the antagonistic coefficient, quantifying the inhibitory effect of one drug on the absorption rate or degree of another drug. On the contrary, the positive parameter is the promotion coefficient.

To validate the effectiveness and interpretability, a control variable approach (fixed parameter) is employed to conduct numerical simulations. For this purpose, the model parameters for each drug used individually are firstly specified, as shown in Table 1. To enhance the readability of the paper, this subsection only presents the scenario where drug Y influences the absorption process of drug X. The remaining cases, along with detailed explanations, are provided in the Supplementary Materials(S1 File.Supplementary.docx). Software MATLAB 2022 (a) was used in Section Numerical Simulations to simulate and estimate all parameters in the PK model.

To examine the influence of drug Y on the absorption process of drug X, according to the coupled model, parameter $\alpha'_{00}$ is varied, while $\alpha'_{12}$, $\alpha'_{21}$, and $\alpha'_{10}$ are fixed at zero. Therefore, the evolutionary trajectories of drug X amount in each

**Table 1. Model parameters for each drug when used individually.**

| Drug | Initia amounts ($10^3 \cdot$mg) | Rate parameters ($h^{-1}$) | | | |
|---|---|---|---|---|---|
| | | $\alpha_{00}\ (\beta_{00})$ | $\alpha_{12}\ (\beta_{12})$ | $\alpha_{21}\ (\beta_{21})$ | $\alpha_{10}\ (\beta_{10})$ |
| X(Y) | 0.6(1) | 0.5(0.6) | 0.5(0.5) | 0.5(0.4) | 0.5(0.3) |

compartment are shown in Fig 2–5, where the black, red, and blue curves illustrate results for $\alpha'_{00} = 0$, $\alpha'_{00} > 0$ and $\alpha'_{00} < 0$, respectively.

As the parameter $\alpha'_{00}$ increases in Fig 2, the absorption rate of drug X from the gastrointestinal compartment to the central compartment increases, leading to earlier complete absorption. As a result, the red curves show a quicker decrease and reach zero sooner. Because of quicker absorption pushing the peak forward, red curves of Fig 3 exhibit a steeper initial climb as a result of the higher absorption rate. The clearance rate rises in proportion to the amount of drug X in the central compartment, which causes the later phase of the red curves to fall more quickly. The red curves fall below the black curve as a result of improved clearance in the early phase, which lowers the quantity of drug X that remains in the central compartment later on.

Fig 4 shows a similar impact as the parameter keeps increasing. Drug X's amount increase in the central compartment facilitates its transmission to the peripheral compartments, speeding up the red curves' beginning climb and resulting in an earlier peak concentration. While greater early-stage elimination lowers the residual amount of drug X in later stages, causing the red curves to dip below the black curve, enhanced removal results in a steeper decline in the later section

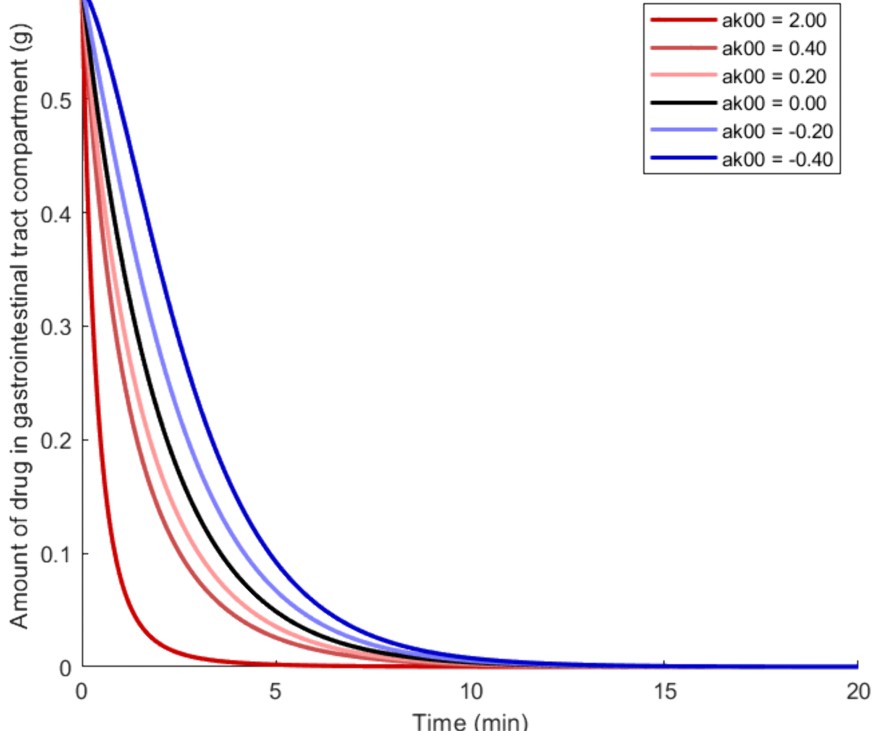

**Fig 2. Amount -time curves of drug X based on the coupled PK model- gastrointestinal tract compartment.**

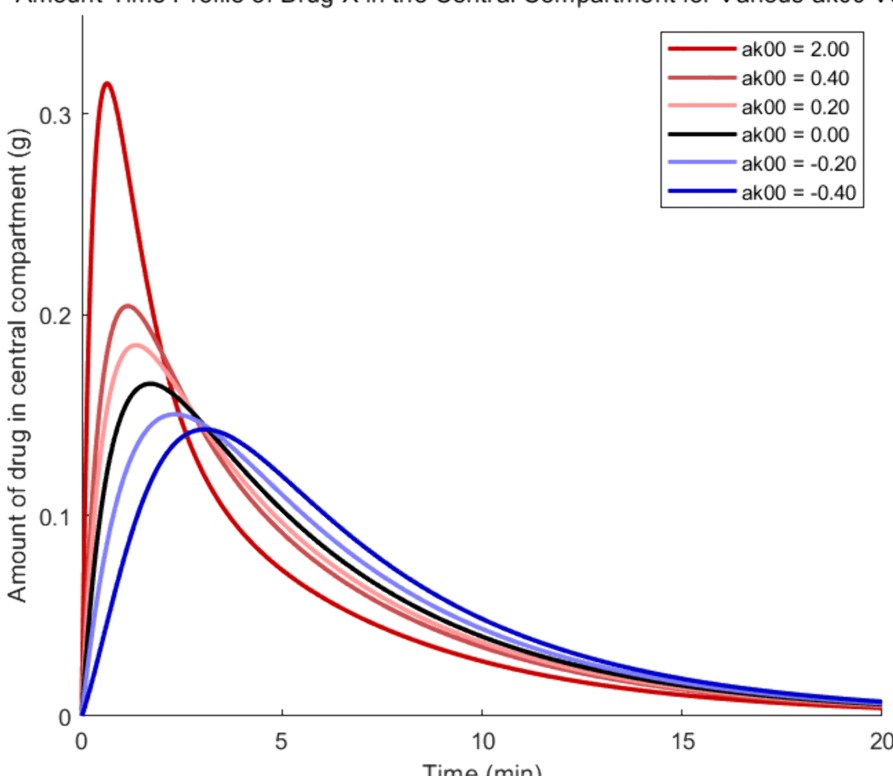

**Fig 3. Amount -time curves of drug X based on the coupled PK model- the central compartment.**

of the red curves. As can be seen in Fig 5, the higher amount of medication X in the center compartment also helps to increase elimination, which raises the red curves' rate of ascent. Furthermore, the inverse of the aforementioned events will transpire if drug Y inhibits drug X's absorption.

Overall, some clear conclusions are obtained from the analysis of parameter variations and their pharmacological implications. The temporal states of plasma drug amount in the four cases provided in the Supplementary Materials are also examined (S1 File.Supplementary.docx,S1-12 Figs, S1-3 Tables).

(1) Effect of absorption changes. The concentration-time peak is shifted to the left and upward with enhanced absorption, allowing effective drug levels in central and peripheral compartments to be reached more quickly, resulting in improved therapeutic efficacy. Conversely, the peak is relocated to the right and downward with reduced absorption, resulting in a slower increase in drug concentration, and potential mitigation of adverse effects.

(2) Effect of transfer from central to peripheral compartment. Increased transfer from central to peripheral compartment allows more of the drug to peripheral tissues, which makes it retained longer in the body and enhances activity at peripheral targets. Reduced this transfer retains more drug in the central compartment, improving efficacy at central sites.

(3) Effect of transfer from peripheral to central compartment. Facilitated transfer from peripheral to central compartments increases drug in the central compartment, enhancing therapeutic effects at central targets. In contrast, reduced this transfer keeps more of the drug in peripheral tissues, which supports prolonged peripheral effects or improving efficacy at those sites.

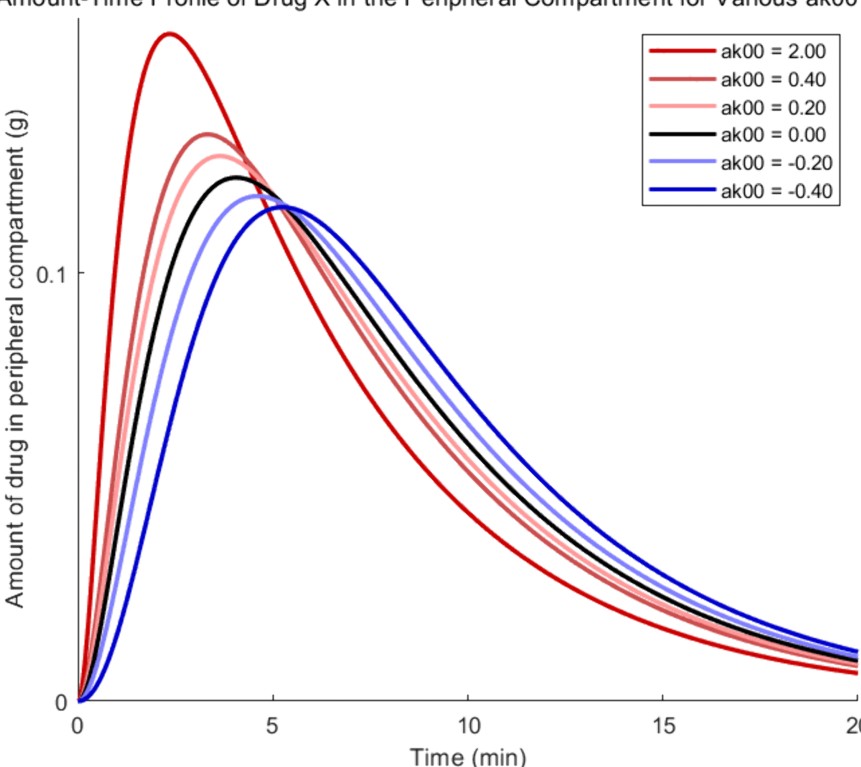

**Fig 4. Amount -time curves of drug X based on the coupled PK model- the peripheral compartment.**

(4) Effect of elimination rates. Faster elimination leads to a reduction in peak drug concentrations, with the peaks being positioned earlier, thereby decreasing the risk of toxicity and minimizing adverse effects. Conversely, inhibited elimination causes the peaks to rise and occur later, extending the drug's presence in the body and amplifying its overall therapeutic potential.

As the absorption parameter $\alpha'_{00}$ gradually increases, the model output results show that the four key pharmacokinetic indicators exhibit consistent trends, as shown in Table 2.

Table 2 shows a significant increase in peak concentration $C_{max}$, a significant advance in peak time $T_{max}$, a slight increase in $AUC$, and a gradual decrease in elimination half-life $t_{1/2}$. Specifically, a higher $\alpha'_{00}$ value means that the rate constant of the absorption process increases with the influence of $y_0$, leading to an accelerated rate of drug entry into the systemic circulation from the absorption site, resulting in a faster increase in blood drug concentration and reaching its peak in a shorter period of time. Due to the constant parameters of the clearance process, the increase in absorption rate has a relatively limited impact on the overall exposure, but it can cause the decline phase to appear earlier and accelerate, resulting in a shortened apparent half-life.

From a clinical perspective, the increase of $\alpha'_{00}$ can be understood as the activation of drug absorption channels or the enhancement of absorption promoting effects, such as the increase of membrane permeability or the reduction of first pass metabolism by excipients, food, or other drugs. This change makes drugs take effect faster and has potential advantages for drugs that require rapid action, such as analgesics, sedatives, or emergency medications. However, an increase in peak concentration also means an increase in instantaneous exposure levels, which may increase the risk of adverse

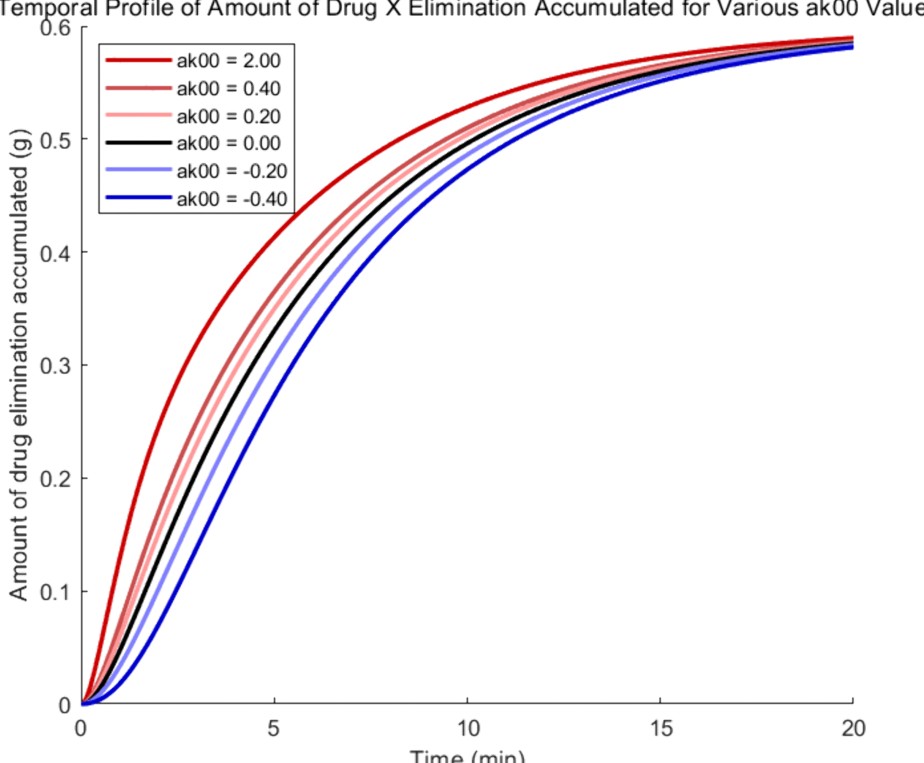

**Fig 5. Amount -time curves of drug X based on the coupled PK model- elimination accumulated.**

**Table 2. Impact of absorption parameters $\alpha'_{00}$ based on a coupled pharmacokinetic model on various pharmacokinetic parameters.**

| Drug | Interaction parameters | Pharmacokinetic parameters | | | |
|---|---|---|---|---|---|
| | $\alpha'_{00}$ (1·10³/(mg·h)) | $C_{max}$ (10³·mg) | $T_{max}$ (h) | *AUC* (10³·mg·h) | $t_{1/2}$ (h) |
| X | −0.4 | 0.14241 | 3.0869 | 1.19917 | 7.9743 |
| | −0.2 | 0.14992 | 2.3374 | 1.19926 | 7.1375 |
| | 0 | 0.16496 | 1.7292 | 1.19933 | 6.1715 |
| | 0.2 | 0.18381 | 1.3703 | 1.19938 | 5.2568 |
| | 0.4 | 0.20294 | 1.1548 | 1.19943 | 4.5160 |
| | 2 | 0.31125 | 0.6399 | 1.19954 | 2.3653 |

reactions associated with peak blood drug levels. In addition, a shortened half-life suggests that the duration of drug efficacy may decrease. If stable blood drug concentration needs to be maintained, adjustments to dosage or frequency may be necessary. Therefore, in prescription design and combination therapy strategies, a balance between absorption rate, onset time of efficacy, and safety should be comprehensively considered to achieve optimal clinical benefits.

## Numerical solution method

To effectively apply the model, accurately estimating its parameters is crucial, which must be derived precisely from experimental data. However, in the model used in this study, the number of parameters can reach up to 16, and the sample size

is limited, posing risks of high parameter non-identifiability and potential parameter correlations. If all drug-related parameters are directly integrated into the modeling process, it would involve 16 parameters, resulting in nonlinear complexity and a large number of parameters. Traditional parameter determination methods may be insufficient to handle this. To address this, the parameter system is divided into two layers: the first layer consists of pharmacokinetic parameters [18] related to individual drugs, estimated independently of experimental data; the second layer comprises interaction parameters for combination therapies. The workflow of these steps is illustrated in Fig 6.

The parameters of a single drug can be directly obtained through experimental data analysis without relying on an optimization process, significantly reducing the dimensionality of subsequent optimization while maintaining physiological interpretability. To account for potential parameter correlations, an appropriate adjustment mechanism is designed in the fitness function during the PSO optimization process. This enables PSO to effectively intensify the optimization of highly correlated parameters, ensuring the independence and accuracy of parameter estimation. This process mitigates error amplification caused by strong correlations, further enhancing model identifiability. This hierarchical strategy not only improves the efficiency and stability of model solving but also helps clarify the hierarchical relationship between monotherapy and combination effects, strengthening the model's interpretability.

To verify the identifiability of the parameters, this study further conducted validation through bootstrap testing and ablation experiments. Bootstrap testing helps assess the model's stability across different data subsets, while ablation experiments test the model's robustness by gradually removing certain parameters or features. These methods ensure that the parameter identification results are reliable and effectively reflect the characteristics of actual drug-drug interactions.

## Optimization process

Estimating the coupling terms during combination medication administration is crucial for figuring out the coupled model parameters because the non-coupled parameters may be solved analytically. Considering the difficulty in solving nonlinear optimization models, this study employs an intelligent optimization algorithm (particle swarm optimization (PSO)) to estimate these parameters.

Particularly, based on the principle of minimum energy and considering the impact of noise errors in experimental data, a regularization strategy is incorporated into the objective function to enhance model complexity and stability. Therefore, the objective function of the optimization process is defined in Eq 3,

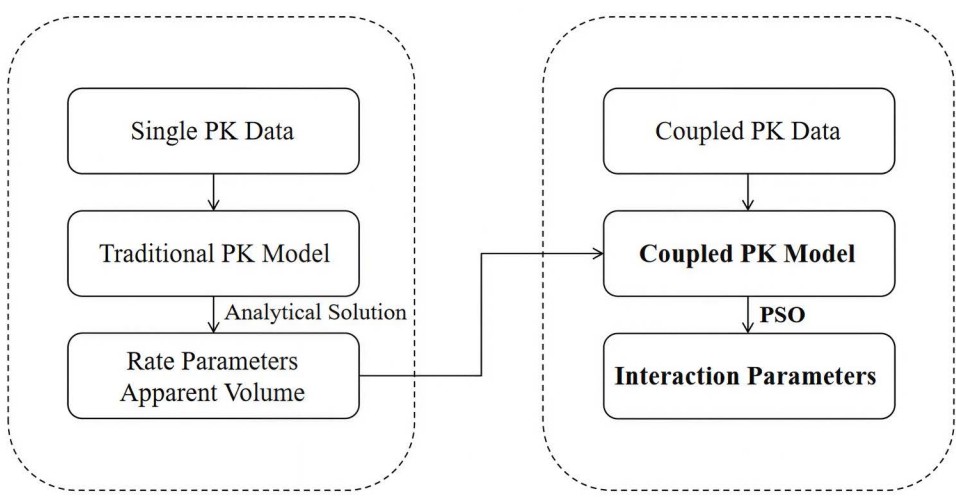

**Fig 6. Flowchart for solving model parameters.**

$$\min z = SSE_x + SSE_y + \lambda \cdot L_2 \tag{3}$$

where

$$SSE_x = \sqrt{\frac{\sum_{i=1}^{n}(x_1^{mod}(t_i) - x_1^{exp}(t_i))^2}{n}}$$

is the mean square error of drug X, and $x_1^{mod}(t_i)$, $x_1^{exp}(t_i)$ are its model data and experiment data at $t = t_i$, respectively.

$$SSE_y = \sqrt{\frac{\sum_{i=1}^{n}(x_2^{mod}(t_i) - x_2^{exp}(t_i))^2}{n}}$$

is the mean square error of drug Y, and $x_2^{mod}(t_i)$, $x_2^{exp}(t_i)$ are its model data and experiment data at $t = t_i$, respectively.

$$L_2 = \sqrt{\alpha_{00}^{'2} + \alpha_{12}^{'2} + \alpha_{21}^{'2} + \alpha_{10}^{'2} + \beta_{00}^{'2} + \beta_{12}^{'2} + \beta_{21}^{'2} + \beta_{10}^{'2}}$$

is the regularization term with the square root of the sum of the squared model parameters. $\lambda$ is the hyperparameter of the regularization strength, better results can be obtained by adjusting $\lambda$ so that $R^2$ is 95–100% when $\lambda=0$.

Coupled PK models transform the parameters in traditional PK models into a linear function. In traditional PK models, parameters must meet the requirement of being positive. Therefore, the values of the interaction parameters in the coupled PK models are constrained by the requirements of traditional PK models; that is Eq 4,

$$\begin{cases} \alpha_{00}' > -\frac{\alpha_{00}}{y_0}, \alpha_{12}' > -\frac{\alpha_{12}}{y_1}, \alpha_{21}' > -\frac{\alpha_{21}}{y_2}, \alpha_{10}' > -\frac{\alpha_{10}}{y_1} \\ \beta_{00}' > -\frac{\beta_{00}}{x_0}, \beta_{12}' > -\frac{\beta_{12}}{x_1}, \beta_{21}' > -\frac{\beta_{21}}{x_2}, \beta_{10}' > -\frac{\beta_{10}}{x_1} \end{cases} \tag{4}$$

where $x_0$, $x_1$ and $x_2$ represent the amount of drug X in the gastrointestinal tract, central compartment, and peripheral compartment, respectively. Similarly, $y_0$, $y_1$ and $y_2$ represent the amount of drug Y in the gastrointestinal tract, central compartment, and peripheral compartment, respectively. The constraints in Eq 4 are strict constraints, but in practical applications, the right side of the inequality should be adjusted to values that can be determined experimentally like Eq 5,

$$\begin{cases} \alpha_{00}' > -\frac{\alpha_{00}}{y_0(0)}, \alpha_{12}' > -\frac{\alpha_{12}}{\max\{y_1^{real}(t)\}}, \alpha_{21}' > -\frac{\alpha_{21}}{y_0(0)}, \alpha_{10}' > -\frac{\alpha_{10}}{\max\{y_1^{real}(t)\}} \\ \beta_{00}' > -\frac{\beta_{00}}{x_0(0)}, \beta_{12}' > -\frac{\beta_{12}}{\max\{x_1^{real}(t)\}}, \beta_{21}' > -\frac{\beta_{21}}{y_0(0)}, \beta_{10}' > -\frac{\beta_{10}}{\max\{x_1^{real}(t)\}} \end{cases} \tag{5}$$

where $x_0(0)$ and $y_0(0)$ are the initial amount of drug X and drug Y, respectively; $\max\{x_1^{real}(t)\}$ and $\max\{y_1^{real}(t)\}$ are the maximum amount of drug X and drug Y in the central compartment, respectively. Since the amount of drug Y in the peripheral compartment $y_2$ cannot be collected from experiments, considering that $y_2$ will certainly not exceed its initial dose, $y_0(0)$ is taken as the maximum value of drug Y in the peripheral compartment here. As long as the parameter does not approach the edge of $-\alpha_{12}/y_0(0)$, the constraint can be accepted. Similarly, taking the initial dose $x_0(0)$ as the maximum value for drug X in the peripheral compartment can also generate a similar constraint.

In addition, regarding the direct measurement data of the peripheral compartment, the typical distribution range of these parameters in relevant literature was also referred to, and reasonable parameter boundary conditions were set in the fitting to ensure the stability of the model and the rationality of the results. These parameters are based on existing

literature and theoretical support, and can provide valuable preliminary estimates for understanding the distribution of drugs in the body.

Subsequently, the constraints for the optimization process are determined. On one hand, since the coupled terms in the model represent various rates, the following inequality Eq 6 holds,

$$AW + U \geq 0 \tag{6}$$

where matrix $A$ is a diagonal matrix with vector $[y_0, y_1, y_2, y_1, x_0, x_1, x_2, x_1]$ along the diagonal and all other elements equal to zero, $U = [\alpha_{00}, \alpha_{12}, \alpha_{21}, \alpha_{10}, \beta_{00}, \beta_{12}, \beta_{21}, \beta_{10}]^T$ is the uncoupling parameter vector, and $W = [\alpha'_{00}, \alpha'_{12}, \alpha'_{21}, \alpha'_{10}, \beta'_{00}, \beta'_{12}, \beta'_{21}, \beta'_{10}]^T$ is the decision variable vector.

On the other hand, some data in the coefficient matrix $A$ are unavailable. Therefore, a scaling method is employed for estimation. For instance, drug concentration data in the peripheral compartment are not reflected in the experimental data, and thus are substituted with the initial amount in the gastrointestinal tract compartment. Therefore, the coefficient matrix A is transformed into

$$[y_0(0), \max\{y_1^{real}(t)\}, y_0(0), \max\{y_1^{real}(t)\}, x_0(0), \max\{x_1^{real}(t)\}, x_0(0), \max\{x_1^{real}(t)\}]$$

Where $x_0(0)$ and $y_0(0)$ are the initial amount of drug X and drug Y, respectively; $\max\{x_1^{real}(t)\}$ and $\max\{y_1^{real}(t)\}$ are the maximum amount of drug X and drug Y in the experiment, respectively.

## Numerical examples

To validate the feasibility and effectiveness of the above numerical solution process, this subsection conducts an optimization process based on the numerical simulation data in Section 2.2. Similarly, to enhance the readability of the paper, this subsection only considers the case described in Section 2.2, while other ones are provided in the Supplementary Materials (S1 File.Supplementary.docx,S4 Table). Due to the individual variability in experimental data, the theoretical values in Table 3 are used to simulate the change of the amount of drug in vivo, and Gaussian noise $N(0, 0.1)$ is added according to the amount of drug at the corresponding time to generate 100 sets of noisy samples. Consequently, the mean and standard deviation of the results of the numerical solution are obtained, as shown in Table 3, with the fitting performance illustrated in Fig 7.

The model fitting results, along with detailed discussions in the Supplementary Materials (S1 File. Supplementary. docx,S13-15 Figs), demonstrate that the hierarchical optimization technique proposed in this study is highly robust and yields satisfactory outcomes. As a result, this model and its numerical solution algorithm can explain the effects of drugs on rates in different compartments and also account for dynamic noise in PK processes. Therefore, the coupled PK model provides deeper insights than the traditional compartmental model, making it more suitable for real-world PK analysis.

**Table 3. *Comparison* of theoretical values and numerical solutions.**

| Interaction parameters (1·10³/(mg·h))(95% CI) | | | |
|---|---|---|---|
| **Drug** | **Interaction parameters** | **Theoretical value** | **Numerical solution** |
| X | $\alpha'_{00}$ | 0.4 | 0.4416 (0.4393, 0.4439) |
| | $\alpha'_{12}$ | 0 | 0.0059 (0.0033, 0.0085) |
| | $\alpha'_{21}$ | 0 | −0.0031 (−0.0031, −0.0031) |
| | $\alpha'_{10}$ | 0 | 0.0072 (0.0055, 0.0089) |

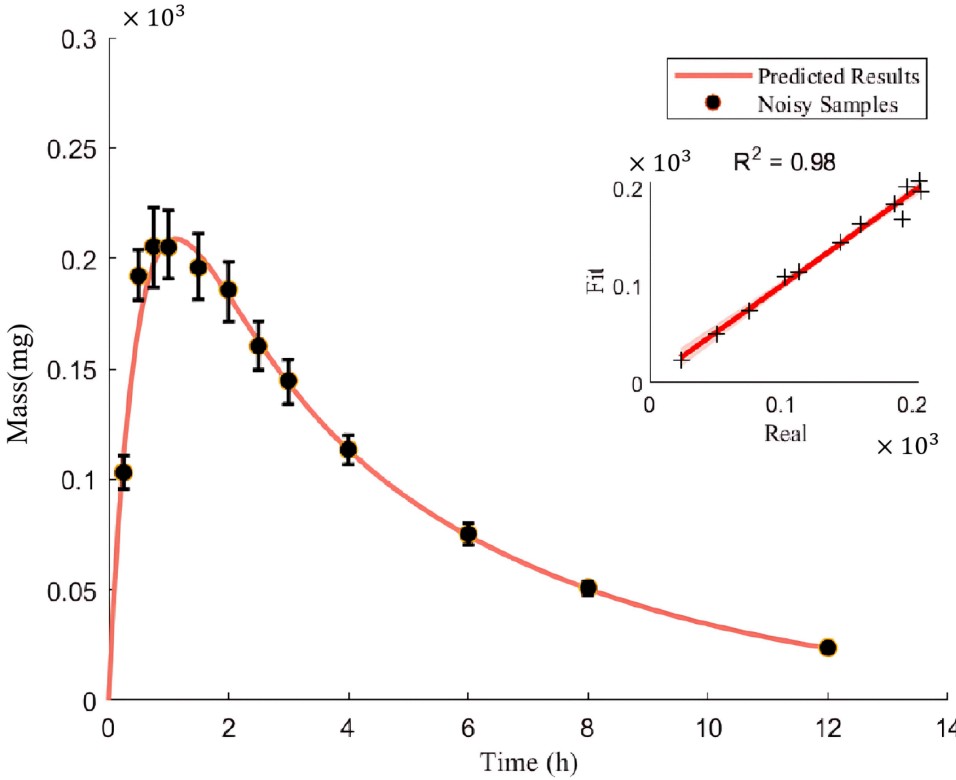

**Fig 7. The fitted curve of drug amount in the central compartment over time under Gaussian noise condition with $\alpha'_{00}$.**

Based on the above results, the model and its numerical solution algorithm can explain the effect of drugs on different compartment rates, and can explain the dynamic noise during PK process. Therefore, the coupled PK model provides deeper insights than traditional compartment models and is more suitable for real-world PK analysis.

## Model performance validation

Although the model has strong explanatory power, further verification is needed to determine whether the selection of regularization parameter λ has sufficient stability during the optimization process. Furthermore, it is worth exploring whether there are other models that achieve the same results with the same accuracy. The dataset used has a small sample size and high heterogeneity ($n = 12$, $n = 16$), which raises doubts about the accuracy of parameter estimation. Therefore, this section 3.3 mainly conducted a systematic verification of the model performance.

Firstly, the optimal value of the regularization parameter λ is determined through sensitivity analysis, and the results of the sensitivity analysis are shown in Table 4. Secondly, from a modeling perspective, explain why methods such as maximum likelihood estimation and Bayesian models are not used. Finally, in order to address the issue of small sample sizes, bootstrap was used for repeated sampling to evaluate the stability and generalization ability of the model on different datasets. To further validate the robustness of the model, this study will compare and analyze the fitting results with noise introduced at fixed time points shown in section 3.2 with the results obtained based on 100 samplings in the absence of noise. The parameter estimation results after multiple samplings are shown in Table 5, Similar to the previous section, in order to improve the readability of the paper, Table 4 only displays the results of four sensitivity analyses. For detailed information, please refer to the Supplementary Materials (S1 File.Supplementary.docx,S5 Table).

**Table 4. Sensitivity analysis of the PSO regularization parameter λ.**

Interaction parameters (1·10³/(mg·h)) (95% CI)

| Drug | Interaction parameters | Theoretical value | PSO regularization parameter λ | | | |
|---|---|---|---|---|---|---|
| | | | 0.1 | 0.3 | 0.6 | 0.9 |
| X | $\alpha'_{00}$ | 0.4 | 0.331 (0.327, 0.335) | 0.377 (0.374, 0.380) | 0.4416 (0.4393,0.4439) | 0.501 (0.498,0.504) |
| | $\alpha'_{12}$ | 0 | 0.0141 (0.0108,0.0174) | 0.0089 (0.0062,0.0116) | 0.0059 (0.0033,0.0085) | 0.0088 (0.0059,0.0117) |
| | $\alpha'_{21}$ | 0 | 0.0085 (0.0079,0.0091) | 0.0052 (0.0047,0.0057) | 0.0031 (0.0031,0.0031) | 0.0020 (0.0014,0.0026) |
| | $\alpha'_{10}$ | 0 | 0.0125 (0.0099,0.0151) | 0.0087 (0.0066,0.0108) | 0.0072 (0.0055,0.0089) | 0.0093 (0.0073,0.0113) |

**Table 5. Results under bootstrap.**

Interaction parameters (1·10³/(mg·h))

| Drug | Interaction parameters | Theoretical value | Numerical solution (95% CI) | P Value |
|---|---|---|---|---|
| X | $\alpha'_{00}$ | 0.4 | 0.40 (0.37,0.43) | 0.634 |
| | $\alpha'_{12}$ | 0 | 0.01 (−0.02,0.04) | 0.056 |
| | $\alpha'_{21}$ | 0 | 0.00 (−0.03, 0.03) | 0.300 |
| | $\alpha'_{10}$ | 0 | 0.00 (−0.03, 0.03) | 0.268 |

In the modeling context of this study (nonlinear differential equation model, lack of analytical solutions, parameter estimation through experimental data), particle swarm optimization (PSO) method is more applicable than maximum likelihood estimation (MLE) and Bayesian inference. The main reason is that such models are usually non convex, non differentiable, or numerically unstable. Traditional MLE relies on a continuously differentiable likelihood function, which is susceptible to the influence of initial values and local extremum. Although Bayesian methods can provide parameter uncertainty assessment, they require repeated numerical integration in high-dimensional parameter spaces, resulting in significant computational overhead. In contrast, PSO, as a global optimization algorithm based on swarm intelligence, does not require gradient information and can directly optimize parameters based on model numerical solutions. It has strong robustness and global convergence ability, making it more efficient and robust in parameter estimation of complex nonlinear dynamic models.

According to the results in Table 4, when the regularization parameter λ is set to 0.6, the model exhibits a relatively stable performance. In addition, after repeated sampling verification using bootstrap method, according to the results in Table 5, the model still maintains high accuracy on different datasets, and there is no significant difference between the numerical solutions of each parameter and the theoretical values, further proving the robustness of the model. Therefore, based on these results, it can be inferred that the proposed model still has good stability and generalization ability when dealing with data with small sample sizes and high heterogeneity.

## Ablation experiment

To verify the influence of different absorption and clearance parameters on the system dynamics process in the model, parameter ablation experiments were conducted in this study. The model contains eight interaction parameters, among which the absorption process (ka) related parameters are $\alpha'_{00}$ and $\beta'_{00}$, the clearance process (CL) related parameters are $\alpha'_{10}$ and $\beta'_{10}$, and the remaining parameters are used to describe the distribution behavior of drugs in the body. Four

parameter setting schemes were constructed to systematically evaluate the contribution of drug interactions to model fitting performance at the absorption and clearance ends:

- **Model 1**: retaining all interaction terms;

- **Model 2**: Fix the absorption end parameter $\alpha'_{00}$ (or $\beta'_{00}$) to 0;

- **Model 3**: Fix the clearing parameter $\alpha'_{10}$ (or $\beta'_{10}$) to 0;

- **Model 4**: Simultaneously fix the absorption end parameter $\alpha'_{00}$ (or $\beta'_{00}$) and the clearing end parameter $\alpha'_{10}$ (or $\beta'_{10}$) to 0;

while keeping the other parameters unchanged,the results are shown in the Fig 8.

As shown in Fig 8, comparing the model fitting goodness ($R^2$) and information criterion (AIC) under four parameter settings, the complete model (Model 1) achieved the highest $R^2$ and the lowest (optimal) AIC, indicating that it is overall optimal in balancing explanatory power and simplicity. In contrast, Model 2 with the removal of the absorption end parameter $\alpha'_{00}$ and Model 3 with the removal of the clearance end parameter $\alpha'_{10}$ both showed a decrease in $R^2$ and an increase in AIC, indicating that both parameters have substantial contributions to the system dynamics behavior; Further removal of both $\alpha'_{00}$ and $\alpha'_{10}$ resulted in the most significant performance degradation in Model 4. The above results unanimously support that preserving the complete absorption and clearance interaction structure can obtain more robust fitting and better information criteria, confirming that the structure setting of Model 1 is the best choice under the current data and model assumptions.

## Model application

In this section, real-world data examples are used to validate the feasibility of the coupled PK model and its solution technique. Meanwhile, a detailed discussion is provided regarding the interpretation of parameter estimates based on the model results.

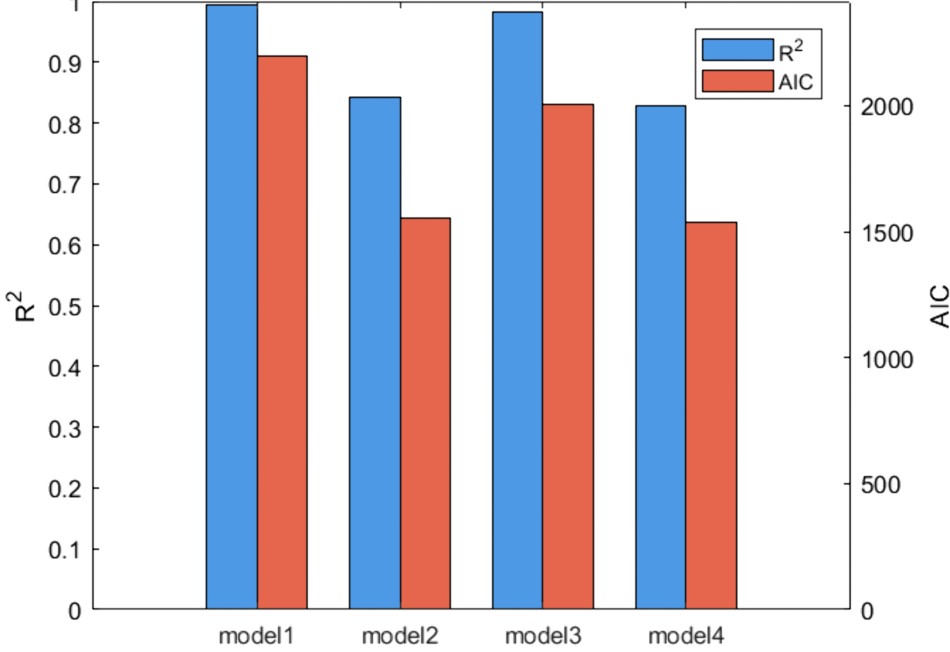

**Fig 8. $R^2$ and *AIC* under different ablation settings.**

In order for the model to more accurately describe the absorption, distribution, metabolism, and excretion processes (ADME) of drugs in different individuals, and to better capture physiological differences and dynamic changes in drug interactions between individuals, the model fitting in this study was based on individual level pharmacokinetic data, rather than aggregated mean data. If average data is used, it may lose information on individual differences, leading to distorted estimates of drug interactions in the model and affecting the final conclusion.

### Data source

Since the hierarchical optimization technique requires experimental data for both single-drug and combined-drug administration, such data are not easily found through a literature search. Therefore, experimental data from the following three studies [19–21] were selected, with specific details provided in Table 6.

The dataset [19] consists of 12 healthy volunteer subjects aged 22–29 years of age, weighing 60–70 kg, and with heights between 170 and 180 cm. A randomized crossover trial method was employed, in which the 12 healthy volunteer subjects were randomly divided into three groups of four individuals each. Each group was sequentially administered Metoprolol 50 mg, Captopril 50 mg, or a combination of both drugs (Metoprolol + Captopril 50 mg), with a 2-week interval between each administration. Blood samples were collected at the following time points: 0.25, 0.5, 0.75, 1.0, 1.25, 1.5, 2.0, 2.5, 3.0, 4.0, 6.0, 8.0, and 12.0 h. Plasma samples containing Captopril were analyzed within 24 h in the dark after centrifugation, while plasma samples containing Metoprolol were stored at −20°C until analysis. Data were collected for both the individual and combined use of Metoprolol and Captopril.

The dataset [20] consists of 16 healthy Caucasian males and females aged 19–55 years with a body amount index (BMI) ranging from 18.5 to 29.9 kg/m$^2$. Imeglimin was administered in two separate instances: as a single dose on Day 1, and in combination with cimetidine on Day 8. From Day 5 to Day 10, 400 mg cimetidine was administered twice daily. Blood samples (6 ml) for Imeglimin determination were collected via venipuncture as follows: pre-dose (0 h) and post-dose at 0.5, 1, 1.5, 2, 3, 4, 6, 8, 10, 12, 24, 36, 48, and 72 h. Blood samples (6 ml) for cimetidine determination were collected as follows: pre-dose (0 h) and pre-morning dose at 0.5, 1, 1.5, 2, 3, 4, 8, and 12 h on Day 8 and Day 9 (24 h), and on Day 10 (48 h). Post-collection, within 1 h, the blood samples were centrifuged at 4°C for 10 minutes at 1500g. Plasma samples were stored at −20°C.

The dataset [21] consists of 16 healthy Caucasian males aged 18–60, with a body mass index (BMI) 20 to 29.9 kg/m$^2$. From Day 1 to Day 6, subjects took Metformin 850 mg tablets and Imeglimin capsules 1500 mg twice daily, and from Day 7 to Day 12, they took Metformin 850 mg tablets and Imeglimin 1500 mg once daily. Plasma sampling for Metformin was collected on Days 4, 5, 6, 10, 11, and 12, while sampling for Imeglimin was performed on Days 10, 11, and 12. For Metformin, blood samples were collected at 1, 1.5, 2, 3, 4, 5, 6, 8, 10, 12, and 24 h post-dose on Day 6 and Day 12, and for Imeglimin, samples were collected at the same time points on Day 12.

In addition, a detailed application of the coupled model and its solution algorithm will be presented for the data [19], while the remaining work is provided in the Supporting Materials(S1 File.Supplementary.docx,S6-9 Tables,S16-18 Figs).

**Table 6. Information of the included studies.**

| Ref | Drug x | Drug y | Main results |
| --- | --- | --- | --- |
| [19] | Metoprolol | Captopril | Captopril increases the blood concentration of met and slows the elimination of metoprolol. Metoprolol has no significant effect on the elimination of captopril. |
| [20] [21] | Imeglimin | Metformin | Imeglimin does not significantly alter the pharmacokinetics of metformin or sitagliptin, and any observed changes in plasma concentrations are not clinically relevant [21]. |

## Model results

Here, metoprolol is used as drug X, and captopril is used as drug Y. Based on the hierarchical optimization technique, the mean and standard deviation of the model rate parameters obtained in the first step from the coupled PK model with zero coupling parameters are shown in Table 7 with the corresponding fitting results in Fig 9, Fig 10.

Subsequently, the second step of the hierarchical optimization technique is conducted using the experimental data from combined drug administration. Based on the objective function Eq 3 and the constraints Eq 4, the mean and standard deviation of the interaction parameters are obtained using the PSO algorithm as shown in Table 8 and the fitting results are illustrated in Fig 11.

A significant correlation between the model's predictions and the experimental findings is demonstrated by the predicted curves, which obtain an $R^2$ of 0.9131 for Met and 0.9341 for Cap, as seen in Fig 11. Deeper understanding

**Table 7. Rate parameters and apparent volume for individual drug administration.**

| $Vc$ (ml) | Rate parameters ($10^{-3} h^{-1}$)(95% CI) | | | |
|---|---|---|---|---|
| | $\alpha_{00}$ | $\alpha_{12}$ | $\alpha_{21}$ | $\alpha_{10}$ |
| 4163.137 (4147.268,4179.006) | 0.205 (0.200,0.210) | 84.192 (82.625,85.759) | 8.122 (7.765,8.479) | 15.162 (14.946,15.378) |
| | $\beta_{00}$ | $\beta_{12}$ | $\beta_{21}$ | $\beta_{10}$ |
| | 0.716 (0.700,0.732) | 69.474 (68.334,70.614) | 19.222 (18.407,20.037) | 14.591 (14.458,14.724) |

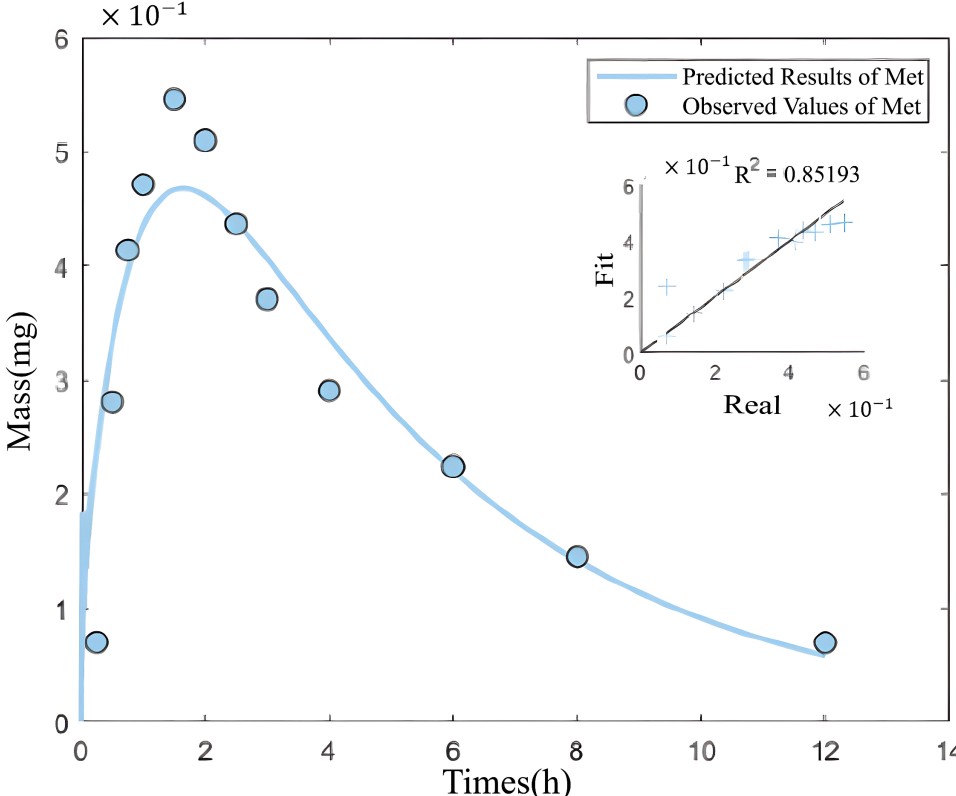

**Fig 9. The fitting curve of drug amount in the central compartment over time under the single-drug condition- metoprolol alone.**

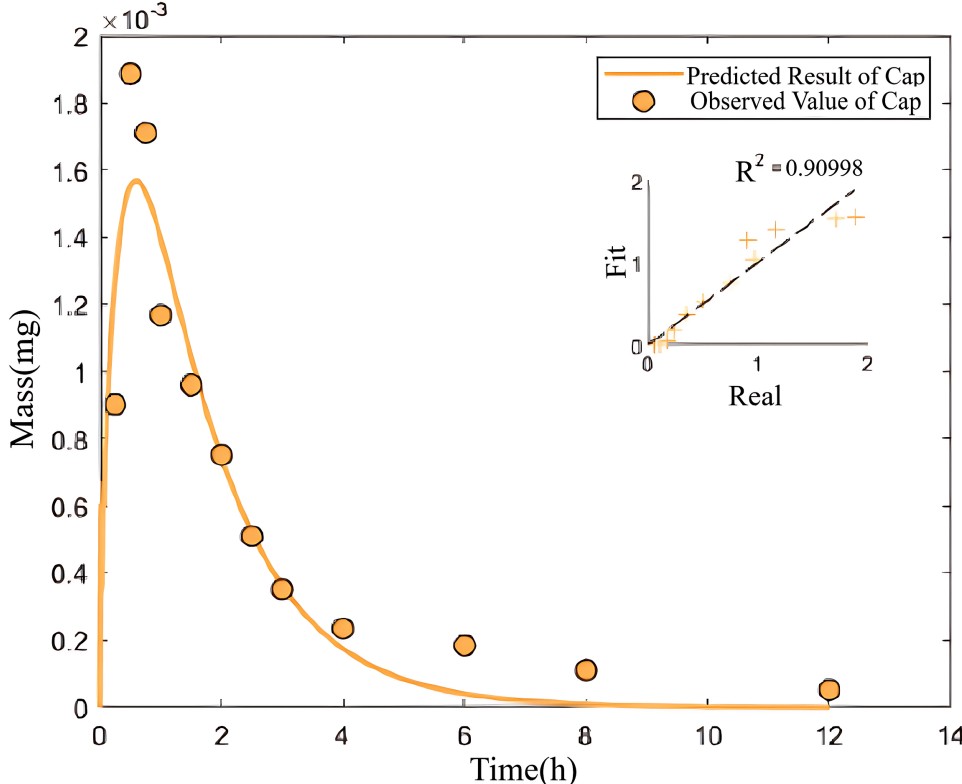

**Fig 10. The fitting curve of drug amount in the central compartment over time under the single-drug condition- captopril alone.**

**Table 8. Interaction parameters for combined drug administration.**

| Interaction parameters (1·10³/(mg·h))(95% CI) | | | |
|---|---|---|---|
| $\alpha'_{00}$ | $\alpha'_{12}$ | $\alpha'_{21}$ | $\alpha'_{10}$ |
| -1.270 (−1.387,-1.153) | −729.2 (−784.5, −673.9) | 585.2 (475.1, 695.3) | −4854 (−5037.1,-4670.9) |
| $\beta'_{00}$ | $\beta'_{12}$ | $\beta'_{21}$ | $\beta'_{10}$ |
| -2.354 (−2.455,-2.253) | −9704 (−10230.4, −9177.6) | 4905 (4725.2,5084.8) | −2481 (−2611.0, −2351.0) |

of the pharmacokinetic interactions between Met and Cap and their implications for therapeutic applications will be provided by the following section, which will explore the coupling parameters and their effects on the two medications.

It is worth mentioning that the parameters solved by the model are slightly larger compared to the results of other studies. But this is a normal phenomenon. The excessively large parameters come from Tables 7,8, where Table 7 shows the parameters of each level under single drug action, and Table 8 shows the interaction parameters under the combined action of two drugs. The reason for the large parameters comes from the dimensions and units. The time unit selected for this study is per hour (h), and the amount of medication transported from one compartment to another per hour is a cumulative process, with larger values being a normal phenomenon. This is the result of mathematical processing and does not affect the biological interpretation of the model itself.

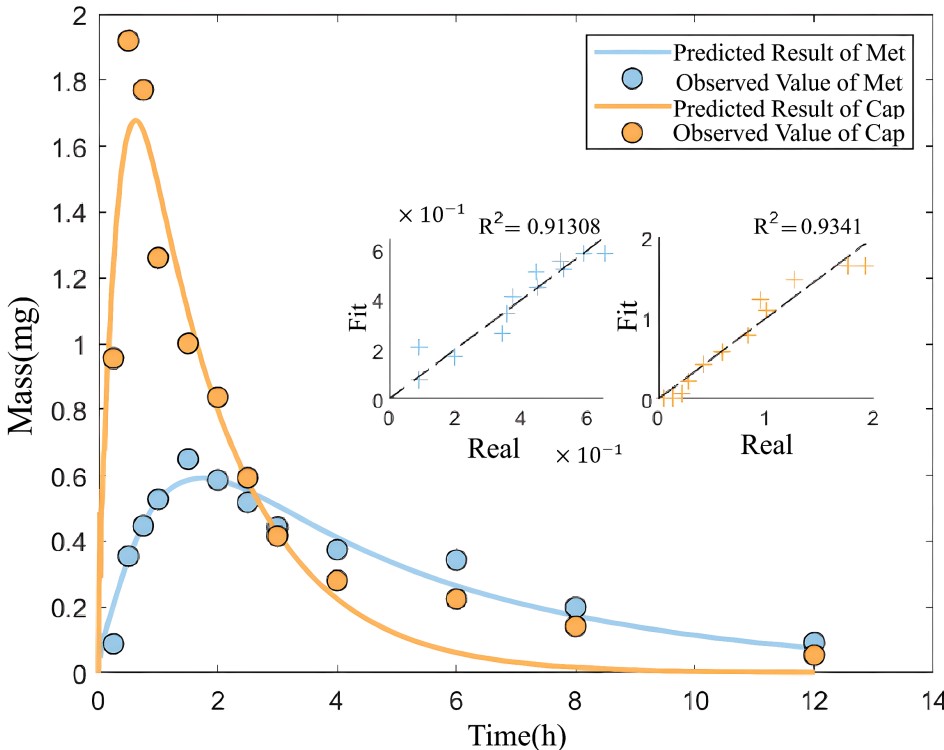

**Fig 11. The fitted curve of drug amount in the central compartment over time under combined-drug conditions.**

## Parameter analysis

The transport of drugs from the central compartment to the peripheral compartment seems to be mutually inhibited by Captopril and Metoprolol, but the transport from the peripheral compartment to the central compartment may be promoted. Due to this dynamic change, it may result in a greater proportion of drugs being retained in the central compartment. The simulation results suggest that Cap and Met may mutually reduce each other's absorption and excretion rates, thereby prolonging the drug's effect in the body.

The pharmacokinetic characteristics of drug X may be affected when used in combination, as the rate parameter of drug X is influenced by the product of its interaction parameter and the amount of drug Y. Specifically, the interaction parameters of drug X are obtained by multiplying the amount of drug Y with its own interaction parameters. The possible effects of these interaction parameters will be discussed in detail below.

### The effect of Cap on Met

The negative value $\alpha'_{00}$ indicates that Cap has an inhibitory influence on Met absorption. The interaction parameter $\alpha'_{00} = -1.270$ indicates that every 1 g of Cap in the gastrointestinal tract compartment reduces the rate of Met absorption by 1.27 $h^{-1}$, but the actual amount of Cap in the gastrointestinal tract compartment is small, making the influence small. Since the maximum amount of Cap in the gastrointestinal tract compartment is 0.05 g of the initial amount, $\alpha'_{00} y_0\left(0\right)/\alpha_{00} \times 100\% = -30.9\%$ and the maximum inhibitory effect on $\alpha_{00}$ is 30.9%. Due to the rapid absorption of Cap from the gastrointestinal tract compartment into the central compartment, this inhibitory influence however decreases rapidly, and its average inhibition of Met absorption by Cap at 2 h, 6 h, and 12 h is 18.08%, 8.27%, and 4.23%, respectively.

The negative value $\alpha'_{12}$ implies that Cap has the influence of inhibiting the transfer of Met from the central to the peripheral compartment. The maximum inhibitory influence on $\alpha_{12}$ is 1.6% and the average inhibition of Met transfer from the central to the peripheral compartment by Cap at 2 h, 6 h and 12 h is 1.07%, 0.53% and 0.27%, respectively.

The negative value $\alpha'_{10}$ means that Cap has an inhibitory influence on the elimination of Met. The maximum suppression influence on $\alpha'_{10}$ is 60.8% and the mean inhibition of Met elimination by Cap at 2 h, 6 h and 12 h is 39.58%, 19.40% and 9.97%, respectively. Meanwhile, a positive value of $\alpha'_{21}$ indicates that Cap has the influence of facilitating the transfer of Met from the peripheral compartment to the central compartment. The interaction parameter $\alpha'_{21} = 585.2$ indicates that each 1 g of Cap in the peripheral compartment increases the rate of Met transfer from the peripheral compartment to the central compartment by 585.2 $h^{-1}$, but the actual amount of Cap in the peripheral compartment is small, making the influence of the effect also small.

It is not possible to estimate the maximum facilitation influence from the experimental data, as the amount data of Cap in the peripheral compartment is not measured in the experiment. However, by simulating the amount prediction of two drugs in the peripheral compartment by the model in this paper, the average facilitation of Met transfer from the peripheral compartment to the central compartment by Cap at 2 h, 6 h and 12 h is 15.78%, 7.41% and 3.84%, respectively.

## The effect of Met on Cap

The negative value $\beta'_{00}$ presents that Met has an inhibitory influence on the absorption of Cap. The maximum inhibitory influence of Met on $\beta'_{00}$ is 16.4%, and the average inhibitory influence at 2 h, 6 h and 12 h is 14.07%, 10.13% and 6.60%, respectively. Meanwhile, the negative value of $\beta'_{12}$ indicated that Met has an inhibitory influence on the transfer of Cap from the central to the peripheral compartments. The maximum inhibitory influence of Met on $\beta'_{12}$ is 9.1%, and the average inhibitory influence on the transfer of Cap from the central to the peripheral compartments at 2 h, 6 h, and 12 h is 6.27%, 5.98%, and 4.05%, respectively.

The negative value $\beta'_{10}$ shows that Met has an inhibitory influence on the elimination of Cap, with a maximum inhibitory influence of 11.1%. The mean inhibition of Cap elimination by Met at 2 h, 6 h and 12 h is 7.63%, 7.28% and 4.92%, respectively. Moreover, a positive value of $\beta'_{21}$ indicated that Cap has the influence of facilitating the transfer of Met from the peripheral to the central compartment. By simulating the amount prediction of two drugs in the peripheral compartment by the model in this paper, the average facilitation of Met transfer from the peripheral compartment to the central compartment by Met at 2 h, 6 h and 12 h is 94.82%, 103.96% and 72.38%, respectively.

Cap and Met seem to mutually inhibit each other's transport from the central compartment to the peripheral compartment, while tending to promote processes in opposite directions. This pattern may partially explain the phenomenon of increased distribution of drugs in the central compartment [22]. This result is somewhat consistent with the pharmacological effects of both drugs [23,24], they both act on vascular receptors, causing vasodilation and lowering blood pressure. Therefore, the simulation results provide a possible mechanism explanation for the pharmacokinetic process, consistent with the known pharmacological characteristics of the two drugs.

Cap may mainly act by affecting the metabolism and excretion of Met, as suggested in reference [19]. Met is primarily metabolized in the liver, with only a small amount excreted in its original form through urine. While Cap is not a typical liver enzyme inhibitor, there is currently no evidence suggesting that Cap is metabolized by cytochrome P450 2D6, nor is the interaction between Cap and Met mediated by this enzyme [25–27]. Therefore, simultaneous administration may interfere with the metabolism of Met, leading to a decrease in its clearance rate and a slight increase in plasma concentration.

On the contrary, the model results show that Met has almost no significant effect on the elimination of Cap. Due to the fact that Cap is metabolized in the liver and mainly excreted through the kidneys, slight liver function regulation usually does not cause significant changes in its pharmacokinetic characteristics. Therefore, when combined with Met, the pharmacokinetic characteristics of Cap remained generally stable.

## Conclusion

This study introduces the hypothesis of parameter heterogeneity and coupling terms based on the traditional atrioventricular model, and establishes a coupled pharmacokinetic (PK) model that can characterize drug interactions. Through numerical simulations, we systematically evaluated the influence of coupling parameters on the dynamic characteristics of plasma concentration time curves under dual drug combination conditions, and proposed a layered optimization strategy to achieve stepwise estimation and stable solution of parameters. This strategy significantly improves the efficiency of parameter identification and the robustness of fitting while maintaining the interpretability of the model structure. Further clinical data validation shows that the proposed model can reasonably reflect the average modulation effect of drug interactions on absorption and clearance processes, providing new ideas for quantitative research on complex drug interaction mechanisms.

Although there are mature tools such as NONMEM and Monolix in the field of group PK modeling that can achieve multi-level parameter estimation and individual difference modeling, the goal of this study is not to replace these systems, but to provide a more concise and flexible optimization and solving framework. This method is particularly suitable for scenarios where data is limited, researchers lack professional modeling platform support, or are in the stage of model exploration. It can serve as a prototype method for drug interaction modeling, providing inspiration and foundation for the construction of more complex systematic models in the future.

It should be pointed out that this research model is based on the assumption that the drug exhibits a linear pharmacokinetic relationship within a specific dose range. This linear assumption can generally reasonably approximate the overall behavior of drug absorption and elimination within the commonly used concentration range in clinical practice. However, when the concentration increases or the transport and metabolic pathways become saturated, the system dynamics may exhibit significant nonlinear characteristics, such as enzyme catalytic saturation, receptor binding saturation, or transmembrane transport limitation, which can cause the model predictions to deviate from the true situation. Therefore, future research should consider introducing nonlinear kinetic parameters (such as the Michaelis constant Km and maximum reaction rate Vmax) to enhance the model's ability to characterize nonlinear absorption and metabolic processes.

The dataset used in this study has a relatively limited sample size and exhibits certain individual heterogeneity. Although a hierarchical estimation and mixed effects framework was used to control for potential confounding factors, it should be acknowledged that the conclusions obtained are only exploratory. The setting of peripheral compartment related parameters is mainly based on the empirical range of apparent distribution volume in literature, and reasonable boundary conditions are set to ensure model stability. Although this method has theoretical and empirical support, due to the lack of direct measurement data, there may still be biases in the relevant estimates, and there is an urgent need for larger sample sizes and multi center datasets to validate and improve the model's universality and extrapolation. These situations will be detailed in the limitations section.

The expansion directions of future research include the following aspects:

Methodology deepening: Introducing Bayesian estimation, regularization terms, or nonlinear dynamic factors on the existing framework to enhance the sensitivity of the model to complex interaction effects and individual differences.

Model system expansion: Based on the atrioventricular model framework, it is further developed into a physiological pharmacokinetic (PBPK) model to achieve more physiologically relevant pharmacokinetic descriptions. The main challenges of PBPK model include: (1) large parameter dimensions and complex transport processes between different organizations; (2) There are significant differences in individual physiological structure, hemodynamics, metabolic enzymes, and transporter activity; (3) The source of parameters relies on literature inference or experimental extrapolation, which has significant uncertainty. Future models can gradually incorporate key physiological and biochemical factors, such as tissue blood flow distribution coefficient, enzymatic reaction parameters, transmembrane transport rate, plasma protein binding rate, liver and kidney clearance pathways, and disease state correction factors, to improve the physiological authenticity and predictability of the model.

Clinical and regulatory significance: The coupled PK model can achieve quantitative evaluation of potential drug interactions in the early stages of drug development, providing basic support for clinical dose optimization and adverse reaction prediction. This framework can also provide auxiliary quantitative tools for drug regulatory agencies to conduct drug interaction risk assessment and label revision in the review process, thereby enhancing the practical application value of the model at the scientific decision-making level.

Model Applicability Extension: The drug-drug interaction model proposed in this study was demonstrated on two pairs of drugs, successfully demonstrating its application in pharmacokinetic analysis. However, in addition to demonstrating these two drug pairs, the model framework has strong generalizability and can be extended to handle other interaction mechanisms, especially drug interactions involving nonlinear processes mediated by transporters or enzymes.

Firstly, the absorption, distribution, metabolism, and excretion (ADME) process of drugs is typically influenced by transport proteins. For example, P-glycoprotein、 Organic anion transporters (OATP) and other transporters play a crucial role in the transmembrane transport of drugs. In this study, although the main focus is on the fundamental kinetic processes between drugs, appropriate extensions to the existing framework can be made to incorporate transporter mediated kinetic equations to describe the interactions between drugs and transporters. Especially at high drug doses, the role of transporters may exhibit saturation kinetics. To handle this nonlinear process, the Michaelis Menten dynamic model can be introduced to further enhance the adaptability of the framework. This extension not only enables the model to handle transporter mediated interactions, but also provides more accurate simulations for drug distribution and excretion processes, thereby enhancing the model's ability to estimate drug drug interaction effects.

In addition to the role of transporters, the metabolic process of drugs is also mediated by enzymes, especially the CYP450 enzyme system in the liver. The activity of these enzymes may change through drug induction or inhibition, leading to nonlinear changes in drug metabolism. In the model framework, although linear drug interactions are currently mainly dealt with, by further expanding the model, enzyme kinetics equations can be introduced to simulate the interactions between drugs and metabolic enzymes. At high concentrations, drugs may trigger enzyme saturation effects, which can be described by the Michaelis Menten kinetics of enzymes to more accurately simulate the metabolic process of drugs. Therefore, this model can adapt to drug interactions involving enzyme mediated nonlinear processes, further improving its ability to describe complex drug metabolism processes.

In the future development of the model, this framework should be further promoted to handle more complex drug drug interaction mechanisms such as immune system mediated drug interactions and the impact of gene polymorphism on drug metabolism. Meanwhile, with the accumulation of more physiological data, the model can be further refined to improve the accuracy of predicting drug interaction mechanisms. This promotion direction not only enhances the application of the model in the field of drug metabolism, but also provides theoretical support for future personalized drug treatment plans.

In summary, the coupled PK modeling framework proposed in this study provides a new quantitative approach for the dynamic interpretation of drug interactions in theory, and demonstrates a feasible path for achieving high robustness modeling under limited data conditions in methodology. Although still limited by linear assumptions, sample size, and parameter simplification, its potential to be extended to PBPK systems will provide more physiologically based prediction and evaluation tools for complex dosing regimens, special population studies, and personalized treatments.

## Limitations

The model developed in this study is based on a fundamental assumption: drugs exhibit linear pharmacokinetic (PK) parameter relationships within a specific dosage range, meaning that drug-drug interactions (DDI) are assumed to follow a linear relationship. By introducing linear coupling terms, the model captures the effects between drugs. This approach simplifies the model structure and computational complexity to some extent, making it particularly suitable for scenarios with low drug concentrations where interactions primarily display approximately linear responses. However, certain drug

interactions in real-world clinical settings may exhibit pronounced nonlinear or saturable pharmacokinetic characteristics. Therefore, under the assumption of solely using linear coupling terms, the model may fail to adequately characterize such nonlinear or saturable drug interactions, especially in high-dose or complex mechanism-of-action scenarios, where its predictive outcomes could deviate from reality. The introduction of the linear assumption primarily stems from a trade-off between model feasibility and interpretability, making it appropriate as an approximate framework for low-dose or relatively simple interaction conditions. For drug combinations known or suspected to involve significant nonlinear/saturable effects, it is necessary to incorporate more complex nonlinear models or real-world clinical data for supplementary analysis and correction.

In addition, the peripheral compartment hypothesis was also used in this study, and an upper bound was established to estimate the peripheral compartment. The upper limit of the peripheral compartment used is conservatively estimated based on existing literature and physiological knowledge, with the aim of limiting the range of drug concentrations in the peripheral compartment. The stability of the model was verified through sensitivity analysis of other key parameters and ablation experiments, and the results showed that the model has a weak dependence on the upper limit value of the peripheral compartment. In other words, within the current set range, changes in the upper limit of peripheral compartments will not significantly alter the main results of pharmacokinetics, especially the estimation of drug drug interaction effects. Therefore, this study did not conduct sensitivity analysis on the upper limit of the hypothesis of the external compartment. Although the setting of upper limits is not sensitive to the results of parameter estimation, it can still introduce errors in certain situations. In future research, it is necessary to further improve the estimation method of peripheral compartments through experimental data or advanced models (such as PBPK models) to enhance the accuracy and stability of the model and ensure more accurate prediction of drug interaction effects.

The small sample size and high heterogeneity of the currently selected dataset are also issues that cannot be ignored. Although potential confounding factors have been controlled through appropriate statistical methods such as stratified analysis and mixed effects models to reduce the impact of heterogeneity on the results, the results of this study are still only preliminary explorations. Additionally, some interaction parameters may be weakly influenced by typical clinical sampling, which is a recognized limitation of this method. Future research should be validated through larger sample sizes of data to further confirm the effectiveness and universality of the model.

In summary, although this study provides a useful preliminary framework for understanding drug interactions, future research still needs to overcome the aforementioned limitations to validate the broad applicability and accuracy of the model.

## Supporting information

**S1 File. Supplementary.** This file contains: **S1 Fig. Amount -time curves of drug X under consideration of parameter** $\alpha_{12}'$ **- gastrointestinal tract compartment. S2 Fig. Amount -time curves of drug X under consideration of parameter** $\alpha_{12}'$ **- the central compartment. S3 Fig. Amount -time curves of drug X under consideration of parameter** $\alpha_{12}'$ **- the peripheral compartment. S4 Fig. Amount -time curves of drug X under consideration of parameter** $\alpha_{12}'$ **- elimination accumulated. S5 Fig. Amount -time curves of drug X under consideration of parameter** $\alpha_{21}'$ **- gastrointestinal tract compartment. S6 Fig. Amount -time curves of drug X under consideration of parameter** $\alpha_{21}'$ **- the central compartment. S7 Fig. Amount -time curves of drug X under consideration of parameter** $\alpha_{21}'$ **- the peripheral compartment. S8 Fig. Amount -time curves of drug X under consideration of parameter** $\alpha_{21}'$ **- elimination accumulated. S9 Fig. Amount -time curves of drug X under consideration of parameter** $\alpha_{10}'$ **- gastrointestinal tract compartment. S10 Fig. Amount -time curves of drug X under consideration of parameter** $\alpha_{10}'$ **- the central compartment. S11 Fig. Amount -time curves of drug X under consideration of parameter** $\alpha_{10}'$ **- the peripheral compartment. S12 Fig. Amount -time curves of drug X under consideration of parameter** $\alpha_{10}'$ **- elimination accumulated. S13 Fig. Noise samples and fitting curves in the central compartment over time-** $\alpha_{12}' = 1$. **S14 Fig. Noise samples and fitting**

curves in the central compartment over time- $\alpha'_{21} = 1$. **S15 Fig. Noise samples and fitting curves in the central compartment over time-** $\alpha'_{10} = 1$. **S16 Fig.The fitted curve of drug amount in the central compartment over time-Imeglimin alone. S17 Fig.The fitted curve of drug amount in the central compartment over time- Metformin alone. S18 Fig.The fitted curve of drug amount in the central compartment over time- oral co-administration. S1 Table. Interaction term parameters** $\alpha'_{10}$ **in a coupled pharmacokinetic model for various pharmacokinetic parameters. S2 Table. Interaction term parameters** $\alpha'_{12}$ **in a coupled pharmacokinetic model for various pharmacokinetic parameters.** S3 Table. Interaction term parameters $\alpha'_{21}$ in a coupled pharmacokinetic model for various pharmacokinetic parameters. S4 Table. Comparison of theoretical values and numerical solutions for drug X. S5 Table. Sensitivity Analysis of the PSO Regularization Parameter $\lambda$. S6 Table. Experimental Data of Metoprolol and Captopril. S7 Table. Experimental Data of Imeglimin and Metformin. S8 Table. The rate parameters by traditional PK modeling. S9 Table. The interaction parameters of coupled PK modeling.
(DOCX)

**S2 File. Raw data.** This file contains: Read me. Sheet 1. Experimental Data of Metoprolol and Captopril. Sheet 2. Experimental Data of Imeglimin and Metformin.
(XLSX)

## Author contributions

**Conceptualization:** Li Yu, Weifeng Jin.

**Data curation:** Hong Huang, Qianqian Chen, Chumeng Zhuang.

**Formal analysis:** Weifeng Jin.

**Methodology:** Hong Huang, Qianqian Chen, Li Yu.

**Software:** Hong Huang, Chaoyang Li.

**Supervision:** Xiaohong Li.

**Validation:** Chumeng Zhuang, Weifeng Jin.

**Visualization:** Hong Huang.

**Writing – review & editing:** Hong Huang, Chaoyang Li, Xiaohong Li.

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
