## [Decision Letter · Decision Letter 0]

17 Sep 2025

Dear Dr. Li,

Thank you for submitting your manuscript to PLOS ONE. After careful consideration, we feel that it has merit but does not fully meet PLOS ONE’s publication criteria as it currently stands. Therefore, we invite you to submit a revised version of the manuscript that addresses the points raised during the review process.

We look forward to receiving your revised manuscript.

Kind regards,

Eshetie Melese Birru, PhD

Academic Editor

PLOS ONE

Journal Requirements:

4. Please include captions for your Supporting Information files at the end of your manuscript, and update any in-text citations to match accordingly. Please see our Supporting Information guidelines for more information: http://journals.plos.org/plosone/s/supporting-information .

Additional Editor Comments:

1. Novelty and Context

The coupled PK model introduces interaction terms and parameter heterogeneity, but the authors should clearly contrast this with PBPK-DDI and nonlinear mixed-effects models that already account for variability and interactions.

Statements such as “existing PK models assume parameters are constants” are oversimplified and ignore nonlinear, time-varying, and interaction models already in use. This weakens the novelty claim unless addressed with updated references.

2. Biological Plausibility of the Model

The model assumes linear relationships between one drug’s mass and the other’s PK parameters. This is questionable because many DDIs involve nonlinear processes (enzyme saturation, transporter inhibition).

Some parameter estimates are extremely large or negative (e.g., −729.2, −9704, 4905, −4854 for interaction coefficients in Table 5), which raises doubts about whether these values are physiologically interpretable or simply mathematical artifacts.

The authors should clarify the clinical meaning of these coefficients and whether constraints should be added to ensure plausibility.

3. Robustness and Validation of the Optimization Method

The hierarchical + PSO approach is novel, but benchmarking is missing. The authors should compare parameter recovery and stability against standard nonlinear regression or mixed-effects methods (NONMEM, Monolix).

It is unclear whether PSO and the regularization term improve estimation robustness in noisy data, or whether simpler maximum likelihood/Bayesian methods would yield comparable results.

A sensitivity analysis of λ (the regularization hyperparameter) is needed — how stable are the fits across values?

4. Use of Real Datasets

The chosen datasets ([19–21]) are small, secondary, and heterogeneous (n=12, n=16). This makes the precision of parameter estimates questionable.

The transformation of plasma concentration to “drug mass” is unconventional, and the justification for this conversion (Eq. 2) needs more explanation.

Some datasets lack peripheral compartment measurements, yet the model estimates peripheral terms. The assumptions used for these missing data should be clearly justified.

5. Interpretation and Clinical Relevance

The model outputs are translated into mechanistic interpretations (e.g., “Captopril inhibits Metoprolol absorption/elimination”), but these statements may overreach the evidence, given the implausible parameter values and data limitations.

Clinical implications (dose adjustment, DDI risk assessment) are only briefly mentioned. The paper would be strengthened by explicitly discussing how this model could support regulatory or therapeutic decision-making, beyond academic modeling.

Reviewers' comments:

Reviewer's Responses to Questions

**Comments to the Author**

1. Is the manuscript technically sound, and do the data support the conclusions?

Reviewer #1: Yes

Reviewer #2: No

2. Has the statistical analysis been performed appropriately and rigorously?

Reviewer #1: Yes

Reviewer #2: No

3. Have the authors made all data underlying the findings in their manuscript fully available?

Reviewer #1: Yes

Reviewer #2: No

4. Is the manuscript presented in an intelligible fashion and written in standard English?

Reviewer #1: Yes

Reviewer #2: Yes

Reviewer #1: The authors have conducted a very interesting, well executed study that advances the field of pharmacokinetic modeling of drug-drug interactions (DDIs). The main novelty here is the coupled model and its hierarchical optimization framework which I feel is a strong contribution to the pharmacometric modeling community.

1. Novel PK model with a clear explanation of how it overcomes the limitations of conventional compartment-based models.

2. Application of the model to real datasets (after robust validation using simulated data with added noise) demonstrate both the validity and robustness of the model

3. Parameter estimation is well explained, and the constraints set are grounded in the feasibility of the underlying experimental design

4. Estimation of interaction parameters and their mechanistic interpretation is compelling and practically relevant to clinicians and pharmacologists.

Suggestions for improvement.

1. Clarity in notation: Mathematical expressions in Eq. 1 and 3 are difficult to follow in the current PDF version due to inconsistencies in notation or poor layout.

2. PBPK expansion: The authors have limited this work to compartmental models, and while they suggest a possible expansion to physiologically based pharmacokinetic (PBPK) modeling, briefly outlining some of the challenges or specific elements of PBPK that they would include would strengthen the discussion.

3. Sensitivity to interaction parameter estimation: Certain terms (e.g. α₁₂′, β₁₂′) are very close to zero and have high variance. Discussion of the significance of these terms to the overall model and possible removal to increase model parsimony would be appropriate if they are not statistically significant.

Reviewer #2: Thank you for the opportunity to review this manuscript. The idea of extending a two-compartment PK framework to capture concentration-dependent drug–drug interactions is interesting and, if refined, could provide a useful middle ground between simple compartmental models and physiologically based PK approaches. The staged estimation process and attempt to apply the model to published clinical data are commendable.

That said, the current version requires major revision before it can be considered for publication. Below I outline specific areas that need attention and provide suggestions for strengthening the work.

1. Accuracy of pharmacological claims

Some statements about known DDIs are not aligned with the published literature. For example, imeglimin has not been shown to prolong metformin half-life or reduce its concentrations in a clinically meaningful way. Similarly, cimetidine–imeglimin interaction studies concluded no significant effect. Metoprolol is CYP2D6-dependent, but captopril is renally excreted and not metabolized through CYP2D6.

Action: Correct these descriptions and ensure all pharmacological claims are supported by current references. Where evidence is limited, keep the interpretation neutral.

2. Model structure, scaling, and units

Some estimated rate constants and interaction terms appear implausibly large when compared with standard PK parameters. It is unclear whether Vc is treated as drug-specific or shared. Units are not always explicit.

Action: Provide unit checks for all parameters, justify parameter magnitudes with literature ranges, and clarify Vc handling. Demonstrate that the model produces realistic predictions at clinical doses expressed in mg.

3. Identifiability and parameter estimation

The model introduces many interaction terms, raising concerns of over-parameterization. There is no evidence that parameters are identifiable given the available data.

Action: Conduct and report identifiability analyses (profile likelihoods, bootstrapping, or Bayesian posteriors). Provide confidence intervals and correlations for parameter estimates. If identifiability is poor, simplify the model structure (for example, restrict interactions to absorption and clearance).

4. Statistical analysis and validation

The choice of λ is subjective, and the evaluation relies almost exclusively on R². More rigorous statistical methods are needed.

Action: Replace heuristic λ selection with a formal criterion such as cross-validation or AIC/BIC. Include diagnostic plots (weighted residuals, visual predictive checks, prediction error metrics). Report uncertainty on all fitted parameters for both synthetic and clinical datasets.

5. Treatment of unobserved compartments

The approach of substituting unknown peripheral masses with maximum observed values from other compartments is ad hoc and could bias results.

Action: Provide a stronger theoretical justification for these constraints or reformulate them in a way that avoids overestimation of interaction effects.

6. Data availability and transparency

The manuscript relies on published data but does not provide digitized datasets or analysis code. The current Data Availability Statement does not meet journal requirements.

Action: Deposit all digitized plasma concentration–time profiles, extraction scripts, and analysis code in a public repository. Update the Data Availability Statement with links to these resources.

7. Clarity and presentation

Equations are duplicated and symbols are sometimes undefined. Figures lack clarity, and the abstract overstates novelty while under-explaining results.

Action:

Number equations sequentially, define every symbol once, and include initial conditions.

Present the ODE system in a clean, consistent format and provide a schematic diagram.

Revise the abstract to state the novel contribution in one sentence, then summarize validation results clearly.

Ensure correct spelling, grammar, and consistent drug abbreviations (e.g., Met, Cap).

8. Mechanistic interpretation

Some mechanistic explanations (e.g., both drugs acting via vasodilation at vascular receptors) are speculative and not supported.

Action: Either remove such statements or support them with high-quality primary references. Focus on describing what the model shows, rather than proposing mechanisms not established in the literature.

9. Additional analyses to strengthen the paper

Sensitivity analysis: Show the effect of each interaction term on Cmax, Tmax, AUC, and half-life.

Model comparison: Test a simpler interaction model (e.g., affecting only CL and ka) and compare using information criteria.

External validation: Apply the model to a second drug pair, even briefly, to demonstrate generalizability.

Robustness: Explore performance under different error structures or sampling schedules.

This manuscript has potential, but to reach publication standard it needs major revisions in five key areas:

1- Enhance the rationale in the introduction by providing more detail on the shortcomings of traditional models.

2- Improve the clarity of the methodology by providing more detailed explanations of the model's equations and justifying the choice of a linear interaction model.

3- Strengthen the discussion by more explicitly connecting the model's findings to their clinical relevance and practical applications.

4- Add a dedicated "Limitations" section to provide a more critical assessment of the study's scope and assumptions.

5- Expand figure captions to make them more descriptive and self-contained.

**Do you want your identity to be public for this peer review?** For information about this choice, including consent withdrawal, please see our Privacy Policy

Reviewer #1: No

Reviewer #2: **Yes:** Faisal Salman Hamad Alshaikh

---

## [Author Response · Author response to Decision Letter 1]

29 Oct 2025

See all replies on "Response to Reviewers".

---

## [Decision Letter · Decision Letter 1]

18 Nov 2025

Dear Dr. Li,

Thank you for submitting your manuscript to PLOS ONE. After careful consideration, we feel that it has merit but does not fully meet PLOS ONE’s publication criteria as it currently stands. Therefore, we invite you to submit a revised version of the manuscript that addresses the points raised during the review process.

The authors have sufficiently addressed the previous comments. Overall, this manuscript presents an innovative coupled pharmacokinetic modeling framework with clear potential for improving the understanding of drug–drug interactions. The approach is well described and supported by both simulation and real-world datasets; however, several scientific aspects would benefit from further clarification to strengthen transparency, interpretability, and physiological justification.

My key comments as Academic Editor are as follows:

Comment 1: Linear interaction assumption

The manuscript relies on linear coupling terms to represent drug–drug interactions. A brief justification of this assumption, given that many DDIs exhibit nonlinear or saturable kinetics, would improve physiological credibility.

Comment 2:  Parameter identifiability

With up to 16 parameters and limited sample sizes, the risk of parameter non-identifiability is high. The authors should comment on potential correlations among parameters and clarify how identifiability was assessed.

Comment 3: Magnitude of interaction parameters

Several estimated interaction coefficients are extremely large relative to conventional PK rate constants. The authors should provide clarification or scaling rationale to support the biological interpretability of these values.

Comment 4: Use of individual vs. mean data

It is unclear whether the model fitting was performed on individual-level data or aggregated means. This distinction is important for understanding variability and the robustness of the interaction estimates.

Comment 5: Peripheral compartment assumptions

Because peripheral masses are not experimentally measured, the model relies on assumed upper bounds. The authors should discuss the implications of these assumptions for parameter accuracy and model stability.

Comment 6: Generalizability of the framework

The proposed model is demonstrated on two drug pairs. A short discussion of how well the framework would generalize to other interaction mechanisms (e.g., transporter- or enzyme-mediated, nonlinear processes) would strengthen the manuscript’s applicability.

We look forward to receiving your revised manuscript.

Kind regards,

Eshetie Melese Birru, PhD

Academic Editor

PLOS ONE

Journal Requirements:

Reviewers' comments:

Reviewer's Responses to Questions

**Comments to the Author**

Reviewer #2: All comments have been addressed

2. Is the manuscript technically sound, and do the data support the conclusions?

Reviewer #2: Yes

3. Has the statistical analysis been performed appropriately and rigorously?

Reviewer #2: Yes

4. Have the authors made all data underlying the findings in their manuscript fully available?

Reviewer #2: Yes

5. Is the manuscript presented in an intelligible fashion and written in standard English?

Reviewer #2: Yes

Reviewer #2: The work is now clearer, more coherent, and far better aligned with the expected standards for pharmacokinetic modelling research. The structure of the paper has improved, the mathematical formulation is easier to follow, and the modelling framework is described with greater transparency. The additions made to the optimisation procedure, the regularisation approach, and the comparison between coupled and uncoupled fits have strengthened the technical presentation.

My assessment of the revision is summarised as follows:

1. Technical soundness and interpretation

The modelling framework is technically sound, and the examples chosen illustrate the behaviour of the coupled PK model effectively. The conclusions are reasonable as long as they are presented as concentration-level interaction patterns rather than mechanistic claims. A small number of pharmacological statements still need to be aligned precisely with the clinical studies cited, particularly for combinations where published evidence shows no clinically meaningful drug–drug interaction.

2. Statistical analysis and robustness

The statistical analysis is much improved and now describes the optimisation process in a more rigorous and reproducible way. A few additions would strengthen the manuscript further:

- Providing confidence intervals or another measure of uncertainty for interaction parameters derived from real clinical datasets.

- Adding a short explanation of practical identifiability, acknowledging which parameters are well informed by the available data and which may be weakly estimated.

- Including basic diagnostic plots such as residuals versus time or predicted versus observed trends, which would complement the fit statistics already presented.

These refinements would finalise the statistical robustness and address the last remaining clarity gaps.

3. Clarity, English usage, and overall presentation

The English is clear and easily comprehensible throughout. A handful of sentences could still be simplified for readability, and terminology should be standardised in places. These are minor edits and do not detract from the overall quality of the revision. Ensuring that all statements about pharmacology are consistent with the cited literature will improve accuracy and avoid overstating interaction mechanisms.

4. Transparency and data availability

To fully meet PLOS transparency requirements, it would greatly benefit readers if the authors provide:

- Machine-readable versions of the concentration–time data used for model fitting.

- The analysis code or scripts required to reproduce the coupled model and optimisation routine.

- An updated Data Availability Statement that explicitly points to these files.

These steps are straightforward and would materially enhance reproducibility.

Below is the specific minor revisions the authors must undertake:

1- Correct remaining pharmacological statements

Ensure all descriptions of the imeglimin–metformin and imeglimin–cimetidine interactions explicitly reflect the findings of the cited clinical studies, which reported no clinically meaningful PK interaction. Adjust any remaining wording that implies reduced metformin concentrations, prolonged half life or enhanced imeglimin exposure.

2- Clarify the metoprolol–captopril description

Confirm that no text suggests captopril is metabolised by CYP2D6 or that the interaction is mediated by CYP pathways. Re-check the Results and Discussion sections to ensure consistency.

3- Provide machine-readable data

Add CSV files containing the exact concentration–time points used for all real clinical datasets. Specify whether they were digitised or extracted from original tables.

4- Provide code or scripts

Upload the model code (for example MATLAB, Python or equivalent) implementing:

- the coupled two-drug system,

- conversion of concentrations to mass, and

- the PSO-based optimisation with regularisation.

Update the Data Availability Statement accordingly.

6- Report parameter uncertainty

Provide confidence intervals, bootstrap results or another measure of uncertainty for the estimated interaction parameters in the real clinical datasets.

7- Add minimal diagnostic plots

Include at least one residual plot (residuals vs time or predicted vs observed) per drug in the Supplementary Materials to demonstrate that no systematic bias is present.

8- Clarify mapping of Drug X and Drug Y

Explicitly state, in each application, which drug is treated as X and which as Y to avoid ambiguity in the interpretation of interaction parameters.

9- Standardise terminology and simplify wording

Replace inconsistent uses of “chamber” vs “compartment”, and “mass” vs “amount”. Simplify a few long sentences in the Results and Discussion for readability.

10- Briefly address identifiability

Add a short statement in the Discussion acknowledging that some interaction parameters may be weakly informed by typical clinical sampling and that this is a recognised limitation of the approach.

**Do you want your identity to be public for this peer review?** For information about this choice, including consent withdrawal, please see our Privacy Policy

Reviewer #2: No

---

## [Author Response · Author response to Decision Letter 2]

25 Nov 2025

For specific replies, please see "Response to Reviewers"

---

## [Editor Report · Decision Letter 2]

1 Dec 2025

Coupled Pharmacokinetic Model Unveils Drug-Drug Interactions in Plasma Concentration

PONE-D-25-24565R2

Dear Dr. Li,

We’re pleased to inform you that your manuscript has been judged scientifically suitable for publication and will be formally accepted for publication once it meets all outstanding technical requirements.

Kind regards,

Eshetie Melese Birru, PhD

Academic Editor

PLOS ONE
---

## [Editor Report · Acceptance letter]

PONE-D-25-24565R2

PLOS One

Dear Dr. Li,

I'm pleased to inform you that your manuscript has been deemed suitable for publication in PLOS One. Congratulations! Your manuscript is now being handed over to our production team.

Kind regards,

on behalf of

Dr. Eshetie Melese Birru

Academic Editor

PLOS One